# The Impact of SRT2104 on Skeletal Muscle Mitochondrial Function, Redox Biology, and Loss of Muscle Mass in Hindlimb Unloaded Rats

**DOI:** 10.3390/ijms241311135

**Published:** 2023-07-06

**Authors:** Lauren T. Wesolowski, Jessica L. Simons, Pier L. Semanchik, Mariam A. Othman, Joo-Hyun Kim, John M. Lawler, Khaled Y. Kamal, Sarah H. White-Springer

**Affiliations:** 1Department of Animal Science, College of Agriculture and Life Science, Texas A&M University and Texas A&M AgriLife Research, College Station, TX 77843, USA; wesolowski@tamu.edu (L.T.W.); jls97@tamu.edu (J.L.S.); piersemanchik@tamu.edu (P.L.S.); 2Department of Kinesiology & Sport Management, School of Education and Human Development, Texas A&M University, College Station, TX 77843, USA; mariam.othman@tamu.edu (M.A.O.); dhwngusdh@tamu.edu (J.-H.K.); jml2621@tamu.edu (J.M.L.); 3Department of Nutrition, Texas A&M University, College Station, TX 77843, USA

**Keywords:** hindlimb unloading, spaceflight, skeletal muscle, electron transfer system, SRT2104, mitochondria, respiration, oxidative stress, antioxidants

## Abstract

Mechanical unloading during microgravity causes skeletal muscle atrophy and impairs mitochondrial energetics. The elevated production of reactive oxygen species (ROS) by mitochondria and Nox2, coupled with impairment of stress protection (e.g., SIRT1, antioxidant enzymes), contribute to atrophy. We tested the hypothesis that the SIRT1 activator, SRT2104 would rescue unloading-induced mitochondrial dysfunction. Mitochondrial function in rat gastrocnemius and soleus muscles were evaluated under three conditions (10 days): ambulatory control (CON), hindlimb unloaded (HU), and hindlimb-unloaded-treated with SRT2104 (SIRT). Oxidative phosphorylation, electron transfer capacities, H_2_O_2_ production, and oxidative and antioxidant enzymes were quantified using high-resolution respirometry and colorimetry. In the gastrocnemius, (1) integrative (per mg tissue) proton LEAK was lesser in SIRT than in HU or CON; (2) intrinsic (relative to citrate synthase) maximal noncoupled electron transfer capacity (E_CI+II_) was lesser, while complex I-supported oxidative phosphorylation to E_CI+II_ was greater in HU than CON; (3) the contribution of LEAK to E_CI+II_ was greatest, but cytochrome *c* oxidase activity was lowest in HU. In both muscles, H_2_O_2_ production and concentration was greatest in SIRT, as was gastrocnemius superoxide dismutase activity. In the soleus, H_2_O_2_ concentration was greater in HU compared to CON. These results indicate that SRT2104 preserves mitochondrial function in unloaded skeletal muscle, suggesting its potential to support healthy muscle cells in microgravity by promoting necessary energy production in mitochondria.

## 1. Introduction

Disuse and mechanical unloading that occur during spaceflight can result in detrimental skeletal muscle atrophy and defects in mitochondrial structure, metabolism, and function. Specifically, spaceflight and microgravity result in decreased muscle fiber size and altered activity of several metabolic enzymes in the soleus and tibialis muscles [1]. Within fast-twitch fibers, space flight decreased succinate dehydrogenase activity in the subsarcolemmal region, while slow-twitch fibers experienced altered distribution of succinate dehydrogenase activity across the entire fiber [2]. Additionally, simulated microgravity increased the production of reactive oxygen species (ROS) in multiple cell types [3,4] and elevated pro-inflammatory signaling [5,6]. Sources of ROS that have been demonstrated to contribute to unloading-induced muscle atrophy include the mitochondria [7] and NADPH oxidase-2 (Nox2) [8,9]. Indeed, mitochondrial dysfunction has been linked to skeletal muscle wasting with denervation [10,11], unloading [7], and cachexia [12,13,14].

Thus, there is a glaring need for preventative measures to maintain skeletal muscle health in astronauts enduring microgravity during their journeys to space. Skeletal muscles are dynamic tissues that respond to alterations in mechanical loading [15]. Increased oxidative stress, inflammation, and insufficient levels of stress proteins are proposed cellular mechanisms by which unloading causes pathology and impairment in musculoskeletal function. Mitochondrial ROS also appears to contribute to unloading-induced atrophy [7]. Mitochondria have many essential cellular roles, including mitochondria’s well known, essential function in energy production. Oxidative energy production in mitochondria is accomplished through enzymatic pathways in the citric acid cycle and the electron transport chain (ETC). Oxidative enzymes, such as succinate dehydrogenase (SDH), as well as substrate flux through the ETC, are important markers of metabolism adversely affected by unloading and spaceflight [16,17]. In addition, ROS production by electron leakage from the ETC can contribute to elevated oxidative stress during unloading [7,18]. Thus, there is current thought that preservation of mitochondrial function and prevention of ROS leakage could mitigate skeletal muscle atrophy that occurs during spaceflight and disuse [14,19].

Recent evidence has suggested that a pathway involving AMP kinase (AMPK) and siruitin-1 (SIRT1) can be instrumental not only in mitochondrial biogenesis [20,21], but also in the preservation of skeletal muscle mass [22]. Sirtuin 1 (SIRT1) is a NAD^+^ dependent deacetylase that influences the activity of several enzymes and transcriptional regulators [23,24,25,26]. SIRT1 mediates responses to inflammatory, metabolic, and oxidative stressors. Specifically, SIRT1 moderates mitochondrial biogenesis through deacetylation of the master regulator of mitochondrial biogenesis, peroxisome proliferator-activated receptor γ coactivator-1α (PGC-1α) [27,28,29]. Previous studies have identified several beneficial effects of SIRT1 activation on musculoskeletal and systemic health via pharmacological treatments, including improved endurance [30,31]. SIRT1 is also involved in skeletal muscle repair [22] and resveratrol’s anti-atrophic effects in aging skeletal muscle occur via SIRT1 [32]. Loss of SIRT1 appears to play a role in food depravation-induced muscle atrophy [33]. However, the potential mitigating role of SIRT1 in unloading-induced mitochondrial function and muscle atrophy is poorly understood.

SRT2104 is a small molecule SIRT1 agonist that induces SIRT1 and downstream signaling [34]. For example, SIRT1 attenuated age-induced loss of muscle mass in both the soleus and gastrocnemius muscles [31]. In addition, SRT2104 treatment reduced pro-inflammatory mediators and elevated PGC-1α during hindlimb unloading in old mice [31]. Together, these results suggest SRT2104 could improve mitochondrial function and protect against muscle wasting. However, the impact of SRT2104 on skeletal muscle mitochondrial function and ROS production has not been determined with mechanical unloading. Therefore, we tested the hypothesis that SRT2104 would rescue mechanical unloading-induced impairments in skeletal muscle mitochondrial function, modulate redox responses, and protect skeletal muscle (e.g., soleus, gastrocnemius) mass. Fischer-344 rats were divided into the following conditions: ambulatory control (CON), hindlimb unloaded for 10 days (HU), and hindlimb unloaded for 10 days while treated with SRT2104 (SIRT).

## 2. Results

### 2.1. Mitochondrial Enzyme Activities

Citrate synthase (CS) and cytochrome *c* oxidase (CCO) activities were quantified in the soleus and gastrocnemius muscles of all rats as indicators of mitochondrial volume density and function, respectively [35,36].

Within the soleus muscle, CS activity, as well as integrative (relative to mg protein) and intrinsic (relative to CS activity) CCO activities, were unaffected by hindlimb unloading or SRT2104 treatment (Figure 1A,C,E). Within the gastrocnemius muscle, CS activity and integrative CCO activity were similarly unaffected by hindlimb unloading or SRT2104 treatment (Figure 1B,D). However, intrinsic CCO activity was lesser in HU rats compared to CON (*p* = 0.006) and SIRT rats (*p* = 0.0002; Figure 1F).

### 2.2. High Resolution Respirometry

Oxidative phosphorylation (P) and electron transfer (E) capacities were determined via high-resolution respirometry. Respiration data are presented either as integrative (relative to tissue wet weight), intrinsic (relative to CS activity; mitochondrial volume density), or as a ratio of the measure of interest to maximal electron transfer capacity (flux control ratio; FCR). Within the soleus muscle, integrative and intrinsic capacities and FCRs were unaffected by hindlimb unloading or SRT2104 treatment (Appendix A). Conversely, within the gastrocnemius muscle, integrative mitochondrial proton LEAK, a dissipative respiratory state not resulting in biochemical work, was lesser in SIRT rats than HU (*p* = 0.002) or CON rats (*p* = 0.03; Figure 2A). Integrative capacities of oxidative phosphorylation supported by complex I (P_CI_ and P_CIG_), maximal coupled oxidative phosphorylation (P_CI+II_), maximal noncoupled electron transfer (E_CI+II_), and electron transfer supported by complex II only (E_CII_) were unaffected by hindlimb unloading or SRT2104 treatment.

Within the gastrocnemius muscle, intrinsic E_CI+II_ was lesser in HU rats than CON (*p* = 0.04), but there was no difference between HU and SIRT rats (Figure 3E). Intrinsic LEAK, P_CI_, P_CI+II_, and E_CII_ were unaffected by hindlimb unloading or SRT2104 treatment.

Also within the gastrocnemius, the contribution of LEAK to maximal electron transfer (flux control ratio; FCR_LEAK_) was greater in HU rats than CON (*p* = 0.03) or SIRT rats (*p* = 0.002; Figure 4A). Additionally, the FCR for oxidative phosphorylation supported by complex I (FCR_PCI_) was lesser in CON rats than HU rats (*p* = 0.01), but it was not different between CON and SIRT rats. Further, the FCR when glutamate was added as an additional complex I substrate (FCR_PCIG_) was lesser in both CON and SIRT rats compared to HU rats (*p ≤* 0.005). However, the FCR for maximal coupled oxidative phosphorylation (FCR_PCI+II_) and noncoupled electron transfer supported by complex II only (FCR_ECII_) were unaffected by hindlimb unloading or SRT2104.

### 2.3. Reactive Oxygen Species

Within the soleus muscle, the rate of production of the ROS, hydrogen peroxide (H_2_O_2_), in isolated mitochondria was greater in SIRT rats compared to CON (*p <* 0.0001) and HU rats (*p* = 0.0007; Figure 5B). Similarly, SIRT rats had the greatest H_2_O_2_ concentrations (vs. HU: *p* = 0.0005; vs. CON: *p <* 0.0001), with HU having intermediate concentrations, and CON rats having the lowest H_2_O_2_ concentrations (vs. HU: *p* = 0.02; Figure 5D).

Similarly, within the gastrocnemius muscle, H_2_O_2_ production was greater in SIRT rats than CON (*p* = 0.004) and HU rats (*p* = 0.003; Figure 5A). Additionally, H_2_O_2_ concentration tended to be greater in HU than CON rats (*p* = 0.08), while SIRT rats had greater H_2_O_2_ concentration than either CON (*p* = 0.003) or HU rats (*p* = 0.01; Figure 5C).

### 2.4. Antioxidant Enzyme Activities

Due to a lack of sample volume, antioxidant enzyme activities were only investigated in the gastrocnemius muscle. Superoxide dismutase (SOD) activity was greater in HU (*p* = 0.004) and SIRT (*p* = 0.009) rats compared to CON, while glutathione peroxidase (GPx) activity did not differ between treatment groups (Figure 6).

### 2.5. Muscle Weights

The right soleus and left gastrocnemius muscles were lesser in weight in HU rats compared CON (*p ≤* 0.0002) rats, and SIRT was lesser than CON (*p ≤* 0.03) (Figure 7). The right soleus was lesser weight in HU rats compared to SIRT (*p* = 0.0002), and the left gastrocnemius tended to be lesser in weight in HU rats compared to SIRT (*p* = 0.07). The left soleus muscle was lesser in weight in HU rats than in SIRT or in CON (*p ≤* 0.0003).

## 3. Discussion

The primary findings of this study are as follows (1) The SIRT1 agonist SRT2104 reduced integrative proton leak in unloaded gastrocnemius muscles. (2) SRT2104 increased cytochrome *c* oxidase activity compared to the unloaded gastrocnemius, while citrate synthase trended higher in the SIRT soleus. (3) Intrinsic O_2_ flux was increased by SRT2104 and FCR leak was reduced in unloaded gastrocnemius muscles. (4) Interestingly, H_2_O_2_ flux and concentration increased with SRT2104 intervention, concomitant with an increase in superoxide dismutase in the gastrocnemius. (5) Importantly, SRT2014 significantly attenuated skeletal muscle atrophy during unloading. These data indicate that SRT2104 protected mitochondrial function in unloaded skeletal muscle. A discussion of the physiological relevance of our findings follows.

This study provides important new information about mitochondrial function markers in a ground analog for spaceflight. The hypothesis that SIRT1 could protect mitochondrial function, perturbations in redox biology, and thus mitigate skeletal muscle atrophy during unloading could be further tested.

Hindlimb unloading resulted in lesser intrinsic maximal electron transfer capacity in the gastrocnemius muscle compared to control rats. However, rats treated with SRT2104, while undergoing hindlimb unloading did not experience the same deficit in maximal electron transfer capacity as those just hindlimb unloaded. Additionally, hindlimb-unloaded rats also had a lesser contribution of complex I-supported oxidative phosphorylation to maximal electron transfer than control rats and SRT2104-treated rats. Complex I is suggested to be the predominant source of ROS compared to the other complexes of the electron transfer system [37,38]. Interestingly, hindlimb-unloaded rats had a greater contribution of LEAK to maximal electron transfer compared to control and SIRT rats. A greater relative proton leak may be an attempted protective mechanism to mitigate elevated ROS concentrations observed in hindlimb-unloaded rats. High oxidative stress and ROS production may induce greater proton leak, generating a positive feedback loop between ROS and proton leak [39]. However, proton leak across the inner mitochondrial membrane results in a decrease in the proton gradient that is generated by the electron transfer system and less protons being utilized in adenosine triphosphate (ATP) production for cellular energy. This may suggest that a greater proton leak also signifies less efficient energy production in hindlimb unloaded rats.

Correspondingly, in the current study, hindlimb-unloaded rats also had greater H_2_O_2_ concentrations in the soleus muscle and tended to have greater H_2_O_2_ concentrations in the gastrocnemius muscle than control rats. Unexpectedly, rats treated with SRT2104 had greater H_2_O_2_ production and concentration in both the gastrocnemius and soleus muscles. Conversely to hindlimb unloaded rats, SRT2104-treated rats had lower integrative LEAK than both control and hindlimb-unloaded rats. This may indicate that SRT2104-treated rats did not experience the same protective proton leak as hindlimb-unloaded rats. However, in previous research, the levels of the antioxidant enzyme SOD2 increased in the muscles of SRT2104-treated mice [31]. This is potentially an adaptive response to mitigate ROS-induced cellular damage, which may lead to mitochondrial dysfunction. The current study noted greater general SOD activity in rats that were hindlimb-unloaded only and in rats treated with SRT2104 during hindlimb unloading. While increasing SOD activity may be a mechanism to combat unloading-induced ROS production, increased SOD without a compensatory increase in catalase or GPx would permit increased levels of H_2_O_2_. However, GPx activity was not different with either treatment. Given that SOD converts the toxic superoxide (O_2_^.−^) to less toxic H_2_O_2_, while GPx eliminates H_2_O_2_ into nontoxic H_2_O and O_2_, the ramification of trading of O_2_^.−^ for H_2_O_2_ is unresolved in unloaded skeletal muscle. Further, the effect of SRT2014 on cellular glutathione levels that could buffer H_2_O_2_ are unknown. Importantly, the current study did not investigate the influence of SIRT activation on Nox2, another predominant source of ROS that we have reported contributes directly to unlading induced atrophy [8]. 

Unloaded mice had markers of oxidative stress and atrophy of myofibers. Previous research indicated that H_2_O_2_ production in the soleus muscle of mice which were immobilized for 14 days was over 10 pmol·min^−1^·mg^−1^ compared to control mice at around 5 pmol·min^−1^·mg^−1^ [7]. In the current study, despite greater H_2_O_2_ production within rats treated with SRT2104 compared to control rats, production did not exceed 5 pmol·min^−1^·mg^−1^ protein; thus, it may be possible that this elevation in H_2_O_2_ production was not significant enough to perpetuate oxidative damage. Additionally, in the current study, mitochondrial capacities in rats treated with SRT2104 were similar to control rats, while hindlimb-unloaded rats exhibited several differences in mitochondrial functionality. This may suggest that, despite elevated ROS production, SRT2104 preserved mitochondrial capacities during hindlimb unloading, potentially due to protective antioxidant mechanisms.

Unlike the gastrocnemius muscle, the soleus muscle did not exhibit significant changes in mitochondrial oxidative phosphorylation or electron transfer capacities in response to hindlimb unloading or SRT2104 treatment. Previous investigations have reported reduced cross-sectional area, a dramatic decrease in total protein, and a decrease in total succinate dehydrogenase activity (activity × cross-sectional area) of both slow- and fast-twitch fibers in the soleus muscle following spaceflight and microgravity [40]. However, succinate dehydrogenase activity decreased in concert with decreased fiber size. Since the mitochondrial measures in the current study were normalized to tissue weight, protein content, or citrate synthase activity, it is possible that mitochondrial measures decreased proportionally with muscle mass. As such, this could be why there were no significant differences in the mitochondrial measures of the soleus after normalization. In support of this, the current study identified that, within the right soleus, muscle mass was lowest in hindlimb-unloaded rats and greatest in control rats. While, within the left soleus, there was no difference in muscle mass of rats treated with SRT2104 and control rats, but the mass of hindlimb-unloaded rats was lower than both groups.

Within the gastrocnemius, muscle mass was lowest in hindlimb-unloaded rats and greatest in control rats. Mitochondrial dysfunction has previously been linked to muscle disuse atrophy in several ways. Mitochondrial ROS production is thought to be a signaling factor, which is connected to protease activity and myofiber atrophy in immobilized muscle [7]. Additionally, changes in mitochondrial morphology and expression of mitochondrial fission genes play a role in mitochondrial dysfunction and AMPK activation, perpetuating muscle wasting [41]. Intrinsic mitochondrial function and maximal electron transfer capacity were lower in the gastrocnemius of hindlimb-unloaded rats than either SRT2104-treated rats or control rats, which might have contributed to depressed muscle mass in hindlimb-unloaded rats. Significantly, rats that were hindlimb-unloaded but treated with SRT2104 maintained mitochondrial capacities similar to ambulatory control rats and maintained muscle mass greater than that of the hindlimb-unloaded rats without SRT2104 treatment.

Within the soleus and the gastrocnemius muscles, there was no treatment-induced differences in citrate synthase activity. However, previous work using transmission electron microscopy identified greater mitochondrial size in muscles of SRT2104-fed mice, despite no change in citrate synthase activity [31]. Therefore, SRT2104 may also alter mitochondrial size, but this was not identified using the techniques employed in the current study. Importantly, greater mitochondrial size does not always translate to enhanced energy production. Thus, the measurement of oxidative phosphorylation, using techniques such as high-resolution respirometry, provides a more comprehensive evaluation of mitochondrial function.

Prior research has suggested that responses to spaceflight differ between slow- and fast-twitch fiber types and that subsarcolemmal mitochondria and intermyofibrillar mitochondria also respond differently to spaceflight [2]. Specifically, subsarcolemmal mitochondria had decreased state II respiration with hindlimb unloading, but intermyofibrillar mitochondria did not [42]. Within the current study, there was no isolation of fiber types or specific mitochondrial subpopulations. Thus, overall, there were no significant changes in citrate synthase activity or integrative cytochrome *c* oxidase activity, but, potentially, there may have been fiber-type or mitochondria-type specific differences, which were not identified via the methods utilized.

Overall, hindlimb unloading did not appear to significantly impact mitochondrial function in the highly oxidative soleus muscle, but it did alter mitochondrial oxidative phosphorylation and electron transfer capacities in the gastrocnemius muscle. Rats were treated with the SIRT1 activator, SRT2104, while hindlimb-unloaded had gastrocnemius mitochondrial capacities closer to those of ambulatory control rats. These results suggest that SIRT1 activation was able to preserve or restore mitochondrial function under conditions where the muscle is inactive. Ultimately, these results suggest SIRT1 may be beneficial in maintaining healthy myocytes in microgravity through promoting appropriate mitochondrial energy production required for cellular action.

## 4. Materials and Methods

### 4.1. Animal Housing and Management

Four-month-old male Fisher-344 (F344) rats, a common rodent model for mechanical unloading and spaceflight, were purchased from Envigo (F344 rats, Envigo RMS, LLC, Indianapolis, IN, USA) and used in this study. Phenotypic muscle fiber atrophy and fiber-type shifts from slow to fast-twitch in F344 rats are similar to humans [40]. Rats were kept in a 12/12 h daylight cycle, temperature-controlled room, and provided chow and water ad libitum. Animal care met all federal stipulations per NIH statutes (NRC Guide for Care and Use of lab animals, 8th Edition, 2011) and was consistent with the Animal Welfare Act and the Public Health Service Policy, as well as the Humane Care and Use of Laboratory Animals.

### 4.2. Experimental Design and Hindlimb Unloading

Fisher-344 rats were divided into three groups: ambulatory controls (CON, *n* = 6), hindlimb-unloaded for 10 days (HU, *n* = 6), and hindlimb-unloaded + 25 mg/kg/day (I.P. injection) SRT2104 (SIRT, *n* = 4). SRT2104 was purchased from MedChemExpress (Sirt1 activator: CAS No. 1093403-33-8, Cat No. HY-15262, MedChemExpress, Brunswick Township, NJ, USA).

Hindlimb unloading in rodents was conducted with an adaptation of the technique previously described by our laboratory [8,9]. We employed a hindlimb harness to produce unloading, as previously described by Mortreux (2020) [43]. The copper wire harness was overfitted with elastic tubing and moleskin as a pliable cushion. The harness was hoisted using a stainless-steel chain attached to a fishing tackle swivel, which was allowed to move freely on a lubricated brass rod. The hindlimb harness allowed rodents to maneuver around the cage with their hindlimb paws just above the cage bedding.

Hindlimb unloading replicates the physiological effects of bedrest and spaceflight [8,9]. Briefly, rats were anesthetized using a cocktail of ketamine (87.5 mg/kg) and xylazine (12.5 mg/kg). To reverse the anesthetic effect of xylazine, 1 mg for every 10 mg xylazine of Atipamezole (5 mg/mL Conc.) was injected IP. Rats were then fitted with a flexible harness attached to a crosswire through a swivel that allowed ambulation around the cage and free access to food and water. Rats were then elevated at a spinal angle of 40° so that hindfeet were approximately 1 cm off the cage floor. At the end of the 10-day hindlimb unloading period, rats from all groups were sacrificed with sodium pentobarbital (150 mg/kg).

### 4.3. Sample Collection

After animal euthanization, gastrocnemius and soleus muscles were dissected, cleaned in PBS, and weighed. Both the soleus and the red gastrocnemius muscles are considered highly oxidative, with more mitochondria content [44,45,46]. Tissues were aliquoted and stored in three ways: (1) flash frozen in liquid nitrogen and stored at −80 °C until mitochondrial enzyme and antioxidant activity analyses; (2) placed directly into 1 mL ice-cold mitochondrial preservation solution (BIOPS; 10 mM Ca-EGTA buffer, 0.1 µM free calcium, 20 mM imidazole, 20 mM taurine, 50 mM K-MES, 0.5 mM dithiothreitol, 6.56 mM MgCl_2_, 5.77 mM ATP, and 15 mM phosphocreatine; pH 7.1) and stored on ice for same day analysis of mitochondrial capacities; and (3) placed in 1mL ice-cold 1X PBS 1 mM EGTA for same-day mitochondrial isolation and analysis of H_2_O_2_ production. The left soleus was used for mitochondrial analyses, while the right soleus was used for quantification of H_2_O_2_ production. The left gastrocnemius was aliquoted and used for all analyses.

### 4.4. Mitochondrial Enzyme Activities

Citrate synthase (CS) and cytochrome *c* oxidase (CCO) activities were determined as measures of mitochondrial volume density and function, respectively, using kinetic colorimetry, as previously described [47,48]. Frozen muscle tissue was cryopulverized (Spectrum™ Bessman Tissue Pulverizer; Spectrum Laboratories, Inc., Rancho Dominguez, CA, USA) and then sonicated 3 times for 3 s each in chilled sucrose homogenization buffer (20 mM Tris, 40 mM KCl, 2 mM EGTA, 250 mM sucrose) with 1 part 5% detergent (n-Dodecyl β-D-maltoside; Sigma D4641). Samples were centrifuged at 11,000× *g* for 3 min at 0 °C, and the supernatant was stored at −80 °C for future analysis. Enzyme activities were determined using a microplate reader (Synergy H1; BioTek Instruments, Winooski, VT, USA). Citrate synthase activity was determined by measuring the linear rate of reaction of free CoA-SH with DTNB at 412 nm at 37 °C and CCO activity determined by measuring the linear rate of oxidation of fully reduced cytochrome *c* at 550 nm at 37 °C. Both assays utilized 80-fold diluted muscle homogenate and were analyzed in duplicate. Intra-assay and inter-assay coefficients of variation (CVs) for CS activity were 4.3% and 1.9%, respectively. Intra-assay and inter-assay CVs for CCO activity were 3.5% and 2.7%, respectively. Each enzyme activity was then normalized to total protein content, which was determined using the Bradford Protein Assay Kit (Thermo Scientific, Rockford, IL, USA). Cytochrome *c* oxidase activity was further normalized to CS activity (intrinsic), giving a measure of function per mitochondria in the sample [36].

### 4.5. High-Resolution Respirometry

High-resolution respirometry was used to measure mitochondrial capacities. All muscle samples were analyzed within 12 h of collection and remained submerged in ice-cold BIOPS solution on ice or at 4 °C until analysis between collection and analysis. Immediately prior to analysis, samples were permeabilized, as described previously [47]. Permeabilized fibers were then rinsed in mitochondrial respiration solution (Mir05; 110 mM sucrose, 60 mM potassium lactobionate, 0.5 mM EGTA, 3 mM MgCl_2_∙6H_2_O, 20 mM taurine, 10 mM KH_2_PO_4_, 20 mM HEPES, 1 g/L BSA, pH 7.1) on a plate rocker for 10 min at 4 °C. Approximately 1.5 to 2.5 mg (wet weight) of rinsed fibers were then immediately added to each chamber of an Oroboros Oxygraph-2k (O2k; Oroboros, Innsbruck, Austria), containing MiR06 (MiR05 + 280 U/mL catalase) and 20 mM creatine. Chambers were maintained at 37 °C and in hyperoxic conditions (200 to 650 µM O_2_) through addition of 200 mM H_2_O_2_. The previously described [49] substrate-uncoupler-inhibitor titration protocol for this study was as follows: (1) complex I substrates, pyruvate (5 mM), and malate (1 mM) to determine nonphosphorylating proton leak (LEAK); (2) adenosine diphosphate (ADP; 2.5 mM), to quantify complex I-supported P (P_CI_); (3) glutamate (10 mM), an additional complex I substrate (P_CIG_); (4) cytochrome *c* (10 µM), to measure integrity of the outer mitochondrial membrane; (5) the complex II substrate, succinate (10 mM), to measure maximal coupled P (P_CI+II_); (6) uncoupler carbonyl cyanide 3-chlorophenylhydrazone (CCCP, 0.5 µM steps), to attain maximal noncoupled E (E_CI+II_); (7) a complex I inhibitor, rotenone (0.5µM), to measure complex II-supported E (E_CII_); and (8) a complex III inhibitor, antimycin A (2.5 µM), to quantify non-mitochondrial residual O_2_ consumption. All data were normalized to residual O_2_ consumption. Samples were re-analyzed if respiration increased greater than 10%, following the addition of exogenous cytochrome *c* (step 4 in protocol listed above). Respiration data are presented either relative to tissue weight (integrative), CS activity (mitochondrial volume density; intrinsic), or as a ratio of the measure of interest to E_CI+II_ (flux control ratio, FCR).

### 4.6. Reactive Oxygen Species

Fat and connective tissue were mechanically removed, and the muscle was minced into fine pieces then incubated in 0.025% trypsin (MilliporeSigma cat#T4799, Burlington, NJ, USA) and mitochondria were isolated, as previously described [50]. Samples were homogenized in mitochondrial isolation medium (MIM: 0.3 M sucrose, 10 mM HEPES, 1 mM EGTA, pH 7.1) containing 1 mg/mL BSA, on ice [51]. Muscle homogenates were subjected to a first centrifugation step (800× *g*, 10 min, 4 °C), and the supernatant was again centrifuged (12,000× *g*, 10 min, 4 °C). The final pellet (isolated mitochondria) was resuspended in 150 µL MIM, and protein quantification was assessed using the Coomasie Bradford Protein assay (Thermo Scientific). H_2_O_2_ production and concentration were quantified fluorometrically using a commercially available kit (AmplexRed, Catalog #A22188, Thermo Fisher Scientific, Waltham, MA, USA).

### 4.7. Antioxidant Enzyme Activities

Superoxide dismutase and glutathione peroxidase activities were measured using commercially available kits (Cayman Chemical, Ann Arbor, MI, USA). Cryopulverized muscle tissue was homogenized in extraction buffer (0.1 M KH_2_PO_4_, 1 mM EDTA, pH 7.2), sonicated on ice, and centrifuged at 14,000× *g* for 2 min at 0 °C, as previously described [51]. Supernatant was collected and stored at −80 °C until further analysis. For SOD, an 80-fold dilution of the homogenate diluted 1:5 in provided sample buffer was analyzed in triplicate on a single plate with an intra-assay CV of 4%. One unit of SOD activity was defined as the amount of enzyme needed to exhibit 50% dismutation of the superoxide radical. For GPx, a 40-fold dilution of the homogenate was analyzed in triplicate on a single plate, with an intra-assay CV of 3%. Both activities were normalized to protein content, determined using the Coomassie Bradford Protein Assay Kit (Thermo Scientific).

### 4.8. Statistical Analysis

Data were analyzed using PROC MIXED in SAS v.9.4 with the fixed effect of treatment and animal within treatment as a random effect. Muscles were analyzed separately. All data are expressed as estimates of least squares means ± SEM. Significance was declared when *p* ≤ 0.05 and trends when 0.05 ≤ *p* ≤ 0.1.

## Figures and Tables

**Figure 1 ijms-24-11135-f001:**
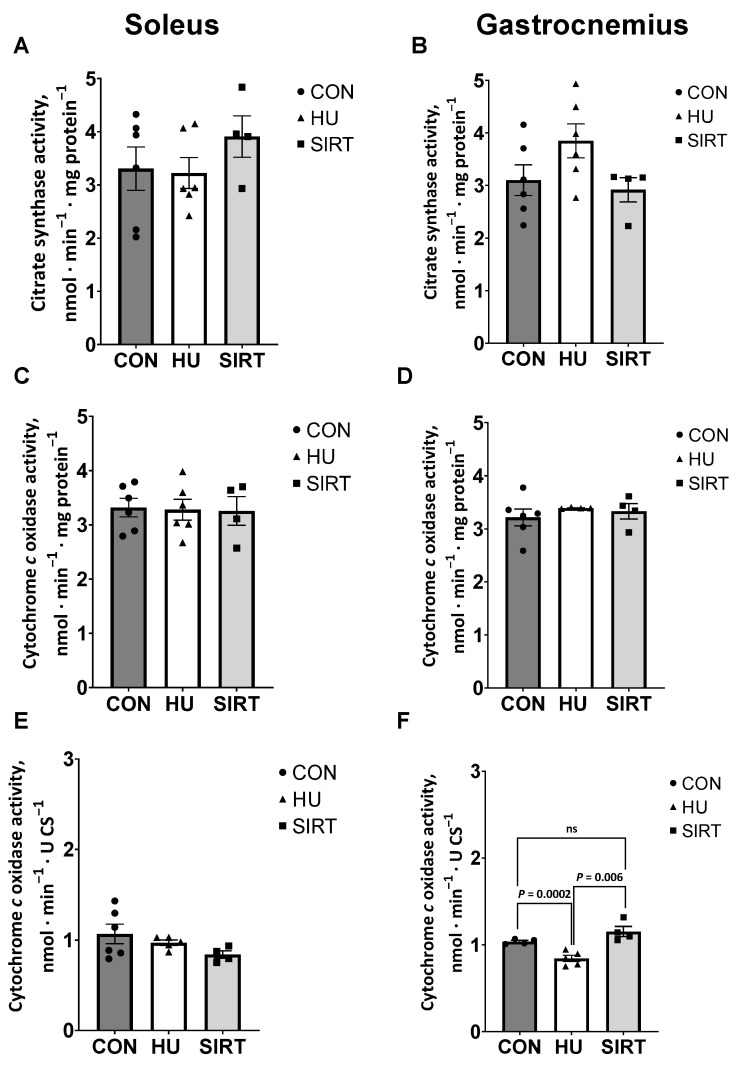
Effect of the Sirtuin-1 Agonist SRT2104 on mitochondrial enzyme activities in hindlimb unloaded rats. (**A**,**B**) Citrate synthase and (**C**,**D**) integrative (relative to mg protein) and (**E**,**F**) intrinsic (relative to CS activity) cytochrome *c* oxidase activities in soleus and gastrocnemius muscles of ambulatory control rats (CON; *n* = 6), rats hindlimb unloaded for 10 days (HU; *n* = 6), and rats hindlimb unloaded for 10 days while treated with SRT2104 (SIRT; *n* = 4). Values are means ± SEM. ns indicates no significant difference (*p* > 0.05).

**Figure 2 ijms-24-11135-f002:**
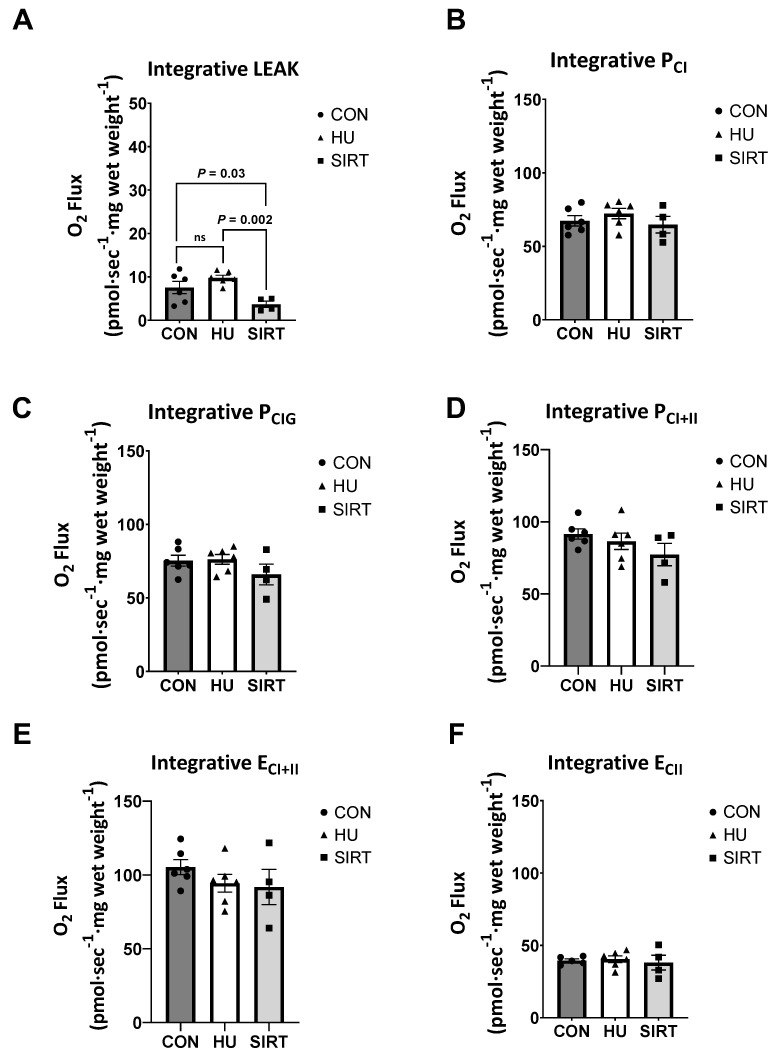
Effect of the Sirtuin-1 Agonist SRT2104 on integrative (relative to tissue wet weight) mitochondrial capacities as measured by high-resolution respirometry in the gastrocnemius muscle of hindlimb unloaded rats. (**A**) Integrative proton LEAK, (**B**,**C**) oxidative phosphorylation supported by complex I (P_CI_ and P_CIG_), (**D**) maximal coupled oxidative phosphorylation (P_CI+II_), (**E**) maximal noncoupled electron transfer (E_CI+II_), and (**F**) electron transfer supported by complex II only (E_CII_) capacities in the gastrocnemius muscle of ambulatory control rats (CON; *n* = 6), rats hindlimb unloaded for 10 days (HU; *n* = 6), and rats hindlimb unloaded for 10 days while treated with SRT2104 (SIRT; *n* = 4). Values are means ± SEM. ns indicates no significant difference (*p* > 0.05).

**Figure 3 ijms-24-11135-f003:**
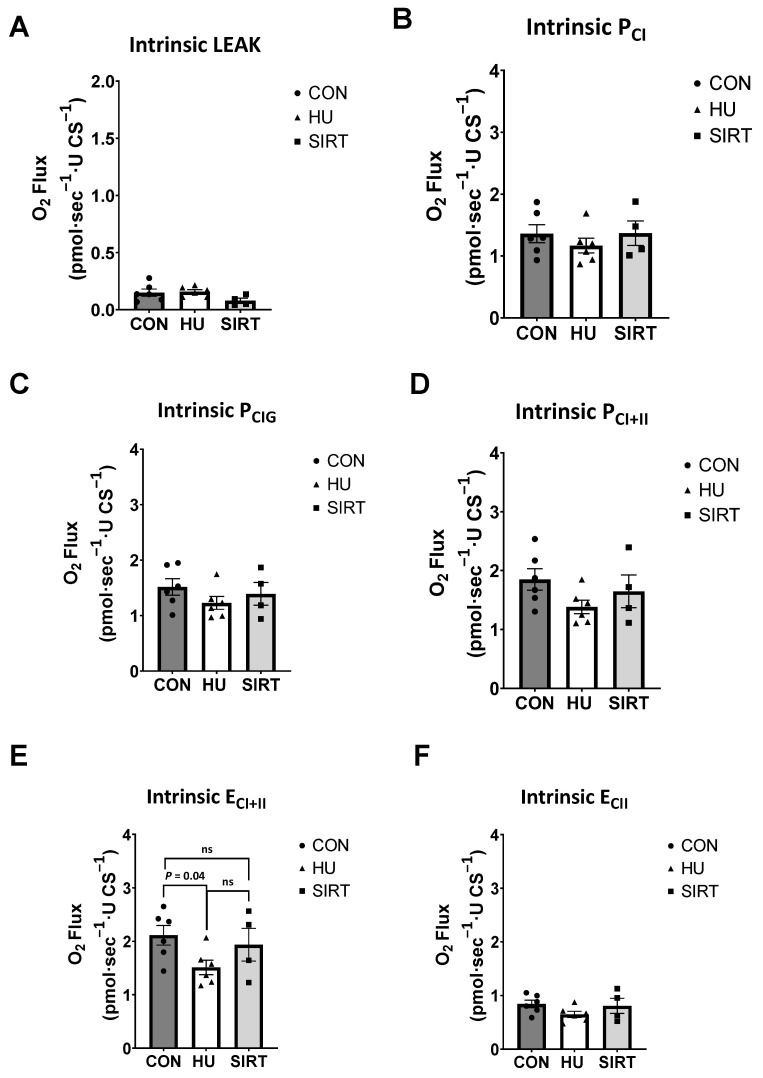
Effect of the Sirtuin-1 Agonist SRT2104 on intrinsic (relative to CS activity) mitochondrial capacities as measured by high-resolution respirometry in the gastrocnemius muscle of hindlimb unloaded rats. (**A**) Intrinsic proton LEAK, (**B**,**C**) oxidative phosphorylation supported by complex I (P_CI_ and P_CIG_), (**D**) maximal coupled oxidative phosphorylation (P_CI+II_), (**E**) maximal noncoupled electron transfer (E_CI+II_), and (**F**) electron transfer supported by complex II only (E_CII_) capacities in the gastrocnemius muscle of ambulatory control rats (CON; *n* = 6), rats hindlimb unloaded for 10 days (HU; *n* = 6), and hindlimb unloaded for 10 days while treated with SRT2104 (SIRT; *n* = 4). Values are means ± SEM. ns indicates no significant difference (*p* > 0.05).

**Figure 4 ijms-24-11135-f004:**
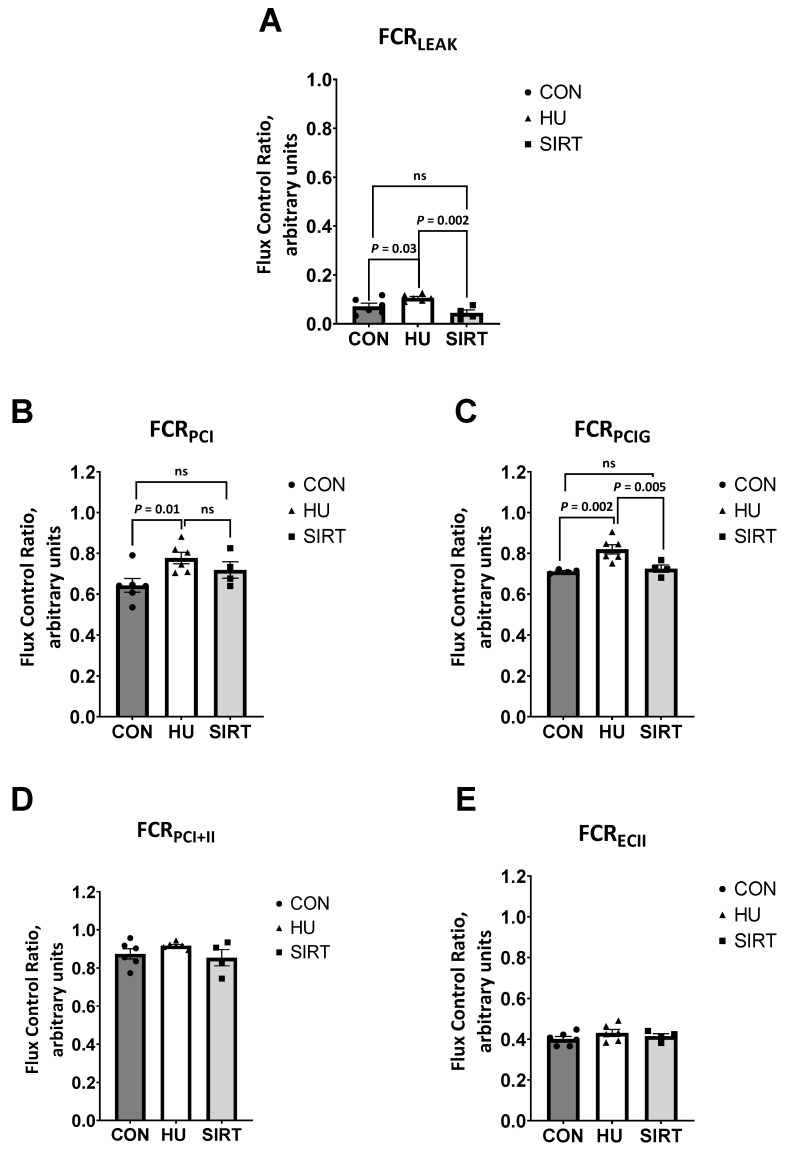
Effect of the Sirtuin-1 Agonist SRT2104 on mitochondrial flux control ratios in the gastrocnemius muscle of hindlimb unloaded rats. (**A**) The ratio of LEAK to maximal electron transfer (flux control ratio; FCR_LEAK_), (**B**) FCR for oxidative phosphorylation supported by complex I (FCR_PCI_), (**C**) FCR when glutamate was added as an additional complex I substrate (FCR_PCIG_), (**D**) FCR for maximal coupled oxidative phosphorylation (FCR_PCI+II_) and (**E**) FCR for noncoupled electron transfer supported by complex II only (FCR_ECII_) of ambulatory control rats (CON; *n* = 6), rats hindlimb unloaded for 10 days (HU; *n* = 6), and rats hindlimb unloaded for 10 days while treated with SRT2104 (SIRT; *n* = 4). Values are means ± SEM. ns indicates no significant difference (*p* > 0.05).

**Figure 5 ijms-24-11135-f005:**
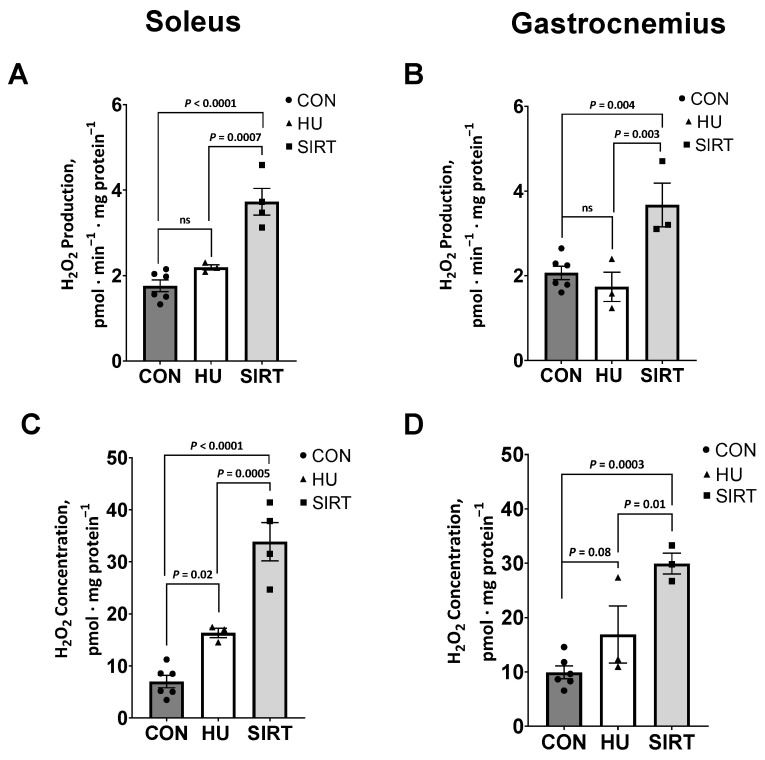
Effect of the Sirtuin-1 Agonist SRT2104 on reactive oxygen species in hindlimb unloaded rats. H_2_O_2_ production and concentration in isolated mitochondria from soleus (**A**,**C**) and gastrocnemius (**B**,**D**) muscles of ambulatory control rats (CON; *n* = 6), rats hindlimbs unloaded for 10 days (HU; *n* = 3), and rats hindlimbs unloaded for 10 days while treated with SRT2104 (SIRT; *n* = 4). Values are means ± SEM. ns indicates no significant difference (*p* > 0.05).

**Figure 6 ijms-24-11135-f006:**
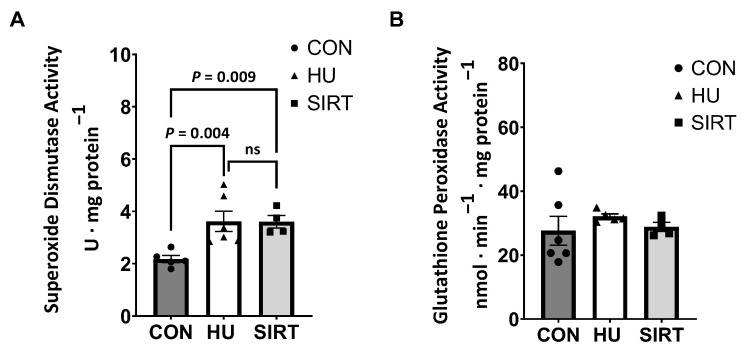
Effect of the Sirtuin-1 Agonist SRT2104 on antioxidant enzyme activities in hindlimb unloaded rats. (**A**) Total superoxide dismutase and (**B**) glutathione peroxidase activities in the gastrocnemius muscle of ambulatory control rats (CON; *n* = 6), rats hindlimbs unloaded for 10 days (HU; *n* = 6), and rats hindlimbs unloaded for 10 days while treated with SRT2104 (SIRT; *n* = 4). Values are means ± SEM. ns indicates no significant difference (*p* > 0.05).

**Figure 7 ijms-24-11135-f007:**
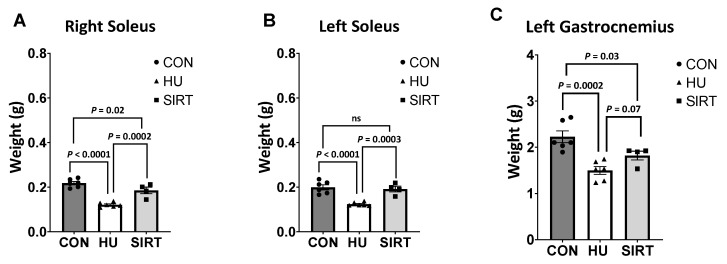
Effect of the Sirtuin-1 Agonist SRT2104 on hindlimb unloaded rats’ muscle weights. Muscle weights of the right soleus (**A**), the left soleus (**B**), and the left gastrocnemius (**C**) of ambulatory control rats (CON; *n* = 6), rats hindlimbs unloaded for 10 days (HU; *n* = 6), and rats hindlimbs unloaded for 10 days while treated with SRT2104 (SIRT; *n* = 4). Values are means ± SEM. ns indicates no significant difference (*p* > 0.05).

## Data Availability

The raw data supporting the conclusions of this article will be made available by the authors upon reasonable request, without undue reservation.

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
