# Peer review of "The Impact of SRT2104 on Skeletal Muscle Mitochondrial Function, Redox Biology, and Loss of Muscle Mass in Hindlimb Unloaded Rats"

_ijms, 2023, doi:10.3390/ijms241311135_

Round 1
Reviewer 1 Report
The Article “The impact of SRT2104 on skeletal muscle mitochondrial function, antioxidant activity, and reactive oxygen species in hindlimb unloaded rats” by Lauren T. Wesolowski et al. is very interesting.
The authors tested the hypothesis that SRT2104 (SIRT1) activator would rescue unloading-induced mitochondrial dysfunction.
1. The conclusions in the Abstract should be expanded.
2. The authors received a lot of data.
However, is not clear in the discussion what this study was conducted for.
In particular, how the obtain results can help to solve the problems associated with skeletal muscle atrophy during spaceflight?
3. In my opinion, the References are insufficient for this study.
The minor aspect:
Please add an extension of all abbreviations in the special section.
Author Response
Author's Reply to the Review Report (Reviewer 1)
The Article “The impact of SRT2104 on skeletal muscle mitochondrial function, antioxidant activity, and reactive oxygen species in hindlimb unloaded rats” by Lauren T. Wesolowski et al. is very interesting.
The authors tested the hypothesis that SRT2104 (SIRT1) activator would rescue unloading-induced mitochondrial dysfunction.
- The conclusions in the Abstract should be expanded.
Although we are limited to 200 words; the Abstract is revised and updated.
- The authors received a lot of data, However, is not clear in the discussion what this study was conducted for. In particular, how the obtain results can help to solve the problems associated with skeletal muscle atrophy during spaceflight?
We have edited substantively the Introduction with careful attention to the narrative and rationale supporting our hypothesis. In addition, the Discussion was edited for clarity and more congruence with the our hypothesis and rationale.
- In my opinion, the References are insufficient for this study.
We thank the reviewer! All the references have been revised and updated.
The minor aspect:
Please add an extension of all abbreviations in the special section.
An Abbreviations section at the end is added as requested.

Reviewer 2 Report
In this study, the Authors investigated the effects of SRT2104, a SIRT1 activator, on mitochondrial function in the gastrocnemius and soleus muscles of rats under mechanical unloading conditions. They compared the results from three groups of rats: mobile control, hindlimb unloading, and hindlimb unloading with SRT2104 treatment. The findings revealed that hindlimb unloading led to mitochondrial dysfunction only in the gastrocnemius muscle. SIRT1 stimulation blocks the negative effects induced by unloading conditions. The Authors suggest that SIRT1 activation may have therapeutic potential in mitigating mitochondrial dysfunction in disuse and microgravity conditions.
The manuscript is well-written and provides new insights into the effect of Sirt1 signaling in muscle. However, some minor revisions are required for it to be published.
1) Regarding the Introduction, I would suggest that the authors expand on the effects of microgravity on muscle by referring to more recent literature. Additionally, they should provide a better explanation for why they chose the gastrocnemius and soleus muscles for their study. Furthermore, I noticed that in line 46, page 2, the two references inserted do not have the same formatting as the others.
2) In the legends of all figures, please indicate to which value of "p" letters a, b, and c correspond, where they are present.
3) In the Discussion, on page 17, lines 217-219, it is reported “In the current study, despite greater in H2O2 production within rats 217 treated with SRT2104 compared to control rats production did not exceed 5 pmol/min/mg protein; thus, it may be possible that this elevation in H2O2 production was not significant enough to perpetuate oxidative damage”. I suggest that the authors perform additional tests to ensure that this increase in ROS does not lead really to oxidative stress. For example, they could measure protein or lipid oxidation markers (such as 3-NT or 4HNE) using Western blotting or investigate whether there is an increase in the expression of the GPX1 or catalase enzymes as an adaptive response.
Author Response
Author's Reply to the Review Report (Reviewer 2)
In this study, the Authors investigated the effects of SRT2104, a SIRT1 activator, on mitochondrial function in the gastrocnemius and soleus muscles of rats under mechanical unloading conditions. They compared the results from three groups of rats: mobile control, hindlimb unloading, and hindlimb unloading with SRT2104 treatment. The findings revealed that hindlimb unloading led to mitochondrial dysfunction only in the gastrocnemius muscle. SIRT1 stimulation blocks the negative effects induced by unloading conditions. The Authors suggest that SIRT1 activation may have therapeutic potential in mitigating mitochondrial dysfunction in disuse and microgravity conditions.
The manuscript is well-written and provides new insights into the effect of Sirt1 signaling in muscle. However, some minor revisions are required for it to be published.
1) Regarding the Introduction, I would suggest that the authors expand on the effects of microgravity on muscle by referring to more recent literature. Additionally, they should provide a better explanation for why they chose the gastrocnemius and soleus muscles for their study. Furthermore, I noticed that in line 46, page 2, the two references inserted do not have the same formatting as the others.
- Introduction:
‘Regarding the Introduction, I would suggest that the authors expand on the effects of microgravity on muscle by referring to more recent literature’
‘Introduction is expanded’
We have edited substantively the Introduction with careful attention to the narrative and rationale supporting our hypothesis. In addition, the Discussion was edited for clarity and more congruence with the our hypothesis and rationale.
- Why GAS and SOLUES?
‘they should provide a better explanation for why they chose the gastrocnemius and soleus muscles for their study’
The gastrocnemius and soleus muscles are more affected by hindlimb unloading due to their role in posture, high proportion of slow-twitch fibers (Type 1; Solues > 85%, Red Gas >50%, (Delp and Duan, 1996)), multi-joint architecture, and functional adaptations. Disuse during unloading leads to atrophy and alterations in these muscle properties (Hord 2021, Lawler 2021).
Furthermore, red gastrocnemius and soleus muscles have a high oxidative capacity and contain more mitochondria due to their role in endurance activities and sustained muscle contractions (Lawler 1993, Criswell, 1992). These muscles primarily consist of slow-twitch fibers, which are rich in mitochondria and well-suited for aerobic energy production. Their abundance of mitochondria allows for efficient ATP generation through oxidative metabolism. Additionally, the muscles' vascularization and higher myoglobin content enhance oxygen supply and facilitate aerobic metabolism (Ratkevičius, 1998). These characteristics enable the gastrocnemius and soleus muscles to sustain prolonged activity, utilize aerobic metabolism effectively, and resist fatigue during endurance tasks (Ono-Moore, 2021).
- References formats:
‘I noticed that in line 46, page 2, the two references inserted do not have the same formatting as the others’
All references throughout the manuscript are reviewed and updated.
2) In the legends of all figures, please indicate to which value of "p" letters a, b, and c correspond, where they are present.
Graphs have been adjusted to include P-values where significant.
3) In the Discussion, on page 17, lines 217-219, it is reported “In the current study, despite greater in H2O2 production within rats 217 treated with SRT2104 compared to control rats production did not exceed 5 pmol/min/mg protein; thus, it may be possible that this elevation in H2O2 production was not significant enough to perpetuate oxidative damage”. I suggest that the authors perform additional tests to ensure that this increase in ROS does not lead really to oxidative stress. For example, they could measure protein or lipid oxidation markers (such as 3-NT or 4HNE) using Western blotting or investigate whether there is an increase in the expression of the GPX1 or catalase enzymes as an adaptive response.
We would like to thank the reviewers for their support and suggestions. In the future experiment, we would extend our analysis to include the suggested parameters, including protein, lipid oxidation markers, .. etc.

Reviewer 3 Report
Review report
Title: The impact of SRT2104 on skeletal muscle mitochondrial function,
antioxidant activity, and reactive oxygen species in hindlimb unloaded rats
The manuscript focuses on the effect of SRT 2104 against mechanical unloading during microgravity leading to skeletal muscle atrophy and impairment of mitochondrial energetics. Overall, the manuscript has been written and is easy to follow. The experimental procedures were described in detail. To prove this hypothesis, the authors determined the oxygen consumption rate by respirometry, electron transfer capacities, H2O2 production, and activities of antioxidant enzymes (SOD and GPx). However, these data are not sufficient to explain the effects of SRT2104 on mitochondrial function and bioenergetics.
Major Comments:
1. Why did the SIRT group have only 4 animals for the experiments?
2. How did the authors confirm SIRT1 activation in the skeletal muscles of the SRT2104 (SIRT1 activator)- treated group? Did the authors performed immunocytochemical (to identify SIRT1 localization) or Western blot analysis (SIRT1 expression)?
3. Superoxide anion radical (O2−.) is the most abundant ROS in mitochondria and is mainly produced by NADPH oxidase (Nox2). Why do the authors focus only on H2O2 concentration in the skeletal muscle mitochondria?
4. In all figures legends, group comparisons (‘a’ and ‘b’) are not mentioned in the manuscript.
5. To derive statistically significant results, 6 animals are necessary. Why does the HU group had only 3 animals for H2O2 analysis in the skeletal muscle (Figure5).
Minor Comments:
1. In Figure 1 legend (Line Number: 90) - Only four animals were used for the mitochondrial enzyme activity assay; however, legends show SIRT; n = 6.
2. In line 295, “ SIRT rats were injected with SRT2104”, is it necessary? Line 294 conveys this meaning.
3. In line 295, “Sirt1 agonists:” - Instead of agonist, activator would be better.
4. Spell out “CV” in the line 340 and 341.
5. Provide an abbreviation for MIM in line number 383.
6. From Line number 362
quantify complex I-supported P (PCI)
complex I substrate (PCIG)
maximal coupled P (PCI+II)
maximal noncoupled E (ECI+II)
complex II-supported E (ECII)
What is P and E represents, and provide details in the manuscript.
Author Response
Author's Reply to the Review Report (Reviewer 3)
The manuscript focuses on the effect of SRT 2104 against mechanical unloading during microgravity leading to skeletal muscle atrophy and impairment of mitochondrial energetics. Overall, the manuscript has been written and is easy to follow. The experimental procedures were described in detail. To prove this hypothesis, the authors determined the oxygen consumption rate by respirometry, electron transfer capacities, H2O2 production, and activities of antioxidant enzymes (SOD and GPx). However, these data are not sufficient to explain the effects of SRT2104 on mitochondrial function and bioenergetics.
Major Comments:
- Why did the SIRT group have only 4 animals for the experiments?
In this experiment, we had limited resources in terms of the SRT2104 drugs, which was enough only to run 4 rats.
- How did the authors confirm SIRT1 activation in the skeletal muscles of the SRT2104 (SIRT1 activator)- treated group? Did the authors performed immunocytochemical (to identify SIRT1 localization) or Western blot analysis (SIRT1 expression)?
SRT2104 is well known as a SIRT1 activator and been used widely. In our experiment, the level of SIRT1 expression was detected by the mean of western blot analysis. Results show a significant upregulation of the SIRT1 levels in the SRT2104 treatments (Kamal et al, 2023).
Kamal KY, et al (2023). The Sirtuin-1 Agonist SRT2104 Mitigates Unloading-Induced Elevation of Inflammatory Markers, Mitochondrial Dysfunction, and Skeletal Muscle Atrophy. The American Society for Gravitational and Space Research meeting, Nov 14 – 18, 2023, Washington, D.C.
- Superoxide anion radical (O2−.) is the most abundant ROS in mitochondria and is mainly produced by NADPH oxidase (Nox2). Why do the authors focus only on H2O2 concentration in the skeletal muscle mitochondria?
‘superoxide has an incredibly short half-life and is difficult to measure. SOD rapidly converts superoxide to H2O2, which is much more stable and easier to quantify’
Additional text acknowledging the role of Nox2 has been added to the discussion (Lines 190-196).
- In all figures legends, group comparisons (‘a’ and ‘b’) are not mentioned in the manuscript.
We thank the reviewer for the comment, the results were revised and we believe that it covers the control vs HU alterations.
- To derive statistically significant results, 6 animals are necessary. Why does the HU group had only 3 animals for H2O2 analysis in the skeletal muscle (Figure5).
We would like to thank the reviewer for that point. Indeed, during the H2O2 analysis, we had a technical issue with the 1st three sample sets, and unfortunately, in this analysis, we lost the samples.
Minor Comments:
- In Figure 1 legend (Line Number: 90) - Only four animals were used for the mitochondrial enzyme activity assay; however, legends show SIRT; n = 6.
Corrected (SIRT; n = 4)
- In line 295, “ SIRT rats were injected with SRT2104”, is it necessary? Line 294 conveys this meaning.
The sentence is updated ‘SRT2104 is purchased from MedChemExpress (Sirt1 activator: CAS No. 1093403-33-8, Cat No. HY-15262, MedChemExpress, USA)’.
- In line 295, “Sirt1 agonists:” - Instead of agonist, activator would be better.
Updated
- Spell out “CV” in the line 340 and 341.
What is the inter-assay CV?
CV has now been defined in the text (line 349). Inter-assay ‘Coefficient of variation’ CV was 1.9% as stated in the text.
- Provide an abbreviation for MIM in line number 383.
As stated in Line 388: MIM refers to the mitochondrial isolation medium.
- From Line number 362, What is P and E represents, and provide details in the manuscript.
quantify complex I-supported P (PCI)
complex I substrate (PCIG)
maximal coupled P (PCI+II)
maximal noncoupled E (ECI+II)
complex II-supported E (ECII)
P and E definitions were added to the text (line 92). ‘Oxidative phosphorylation (P) and electron transfer (E)’

Round 2
Reviewer 3 Report
The authors have carried out the comments and suggestions.